# Novel Fosfomycin Resistance Mechanism in *Pseudomonas entomophila* Due to Atypical Pho Regulon Control of GlpT

**DOI:** 10.3390/antibiotics13111008

**Published:** 2024-10-26

**Authors:** Laura Sánchez-Maroto, Pablo Gella, Alejandro Couce

**Affiliations:** Centro de Biotecnología y Genómica de Plantas (CBGP), Universidad Politécnica de Madrid (UPM), 28223 Madrid, Spain; pablo.gella@upm.es (P.G.)

**Keywords:** fosfomycin resistance, mutation rate, GlpT, Pho regulon, promoter evolution

## Abstract

**Background/Objectives:** *Pseudomonas entomophila* is a ubiquitous bacterium capable of killing insects of different orders and has become a model for host–pathogen studies and a promising tool for biological pest control. In the human pathogen *Pseudomonas aeruginosa*, spontaneous resistance to fosfomycin arises almost exclusively from mutations in the glycerol-3-phosphate transporter (GlpT), the drug’s sole entry route in this species. Here, we investigated whether this specificity is conserved in *P. entomophila*, as it could provide a valuable marker system for studying mutation rates and spectra and for selection in genetic engineering. **Methods:** We isolated 16 independent spontaneous fosfomycin-resistant mutants in *P. entomophila*, and studied the genetic basis of the resistance using a combination of sequencing, phenotyping and computational approaches. **Results:** We only found two mutants without alterations in *glpT* or any of its known regulatory elements. Whole-genome sequencing revealed unique inactivating mutations in *phoU*, a key regulator of the phosphate starvation (Pho) regulon. Computational analyses identified a PhoB binding site in the *glpT* promoter, and experiments showed that *phoU* inactivation reduced *glpT* expression nearly 20-fold. While placing a sugar-phosphate transporter under the Pho regulon may seem advantageous, bioinformatic analysis shows this configuration is atypical among pseudomonads. **Conclusions:** This atypical Pho regulon control of GlpT probably reflects the peculiarities of *P. entomophila*’s habitat and lifestyle; highlighting how readily regulatory evolution can lead to the rapid divergence of resistance mechanisms, even among closely related species.

## 1. Introduction

*Pseudomonas entomophila* is a strictly aerobic, motile, Gram-negative bacterium commonly found in *diverse soil* environments [1]. While first identified in 2005 as a pathogen of the fruit fly [2], *P. entomophila* can kill larvae from various insect orders, including *Diptera*, *Lepidoptera*, and *Coleoptera* [3]. Beyond these entomopathogenic properties, *P. entomophila* can also kill other organisms, such as amoebas, nematodes, and phytopathogenic fungi [3]. Consequently, *P. entomophila* has attracted much interest due to its potential as a biocontrol agent against a range of plant pathogens and insect pests, either as a living organism or through its isolated insecticidal effectors [3,4,5]. In addition to these practical considerations, *P. entomophila* has become an important model organism for studying insect–microbe interactions [6]. Its ability to establish successful infections in laboratory strains of the fruit fly provides an ideal system for exploring various aspects of host–pathogen biology, including the insect’s physiological and immune responses to infection [7,8], the dynamics of release and mechanisms of action of both antimicrobial peptides and insecticidal effectors [2,9], and the capacity of both organisms to acclimatize or even adapt to the challenges imposed by the other [10,11].

To fully realize the potential of *P. entomophila* as a model system, studies exploring whether the knowledge and genetic tools available for other pseudomonads are transferable to this emerging model are much needed. Here, we focus on the mechanisms of intrinsic resistance to the antibiotic fosfomycin, a spontaneous resistance marker system well characterized in *Escherichia coli* and *Pseudomonas aeruginosa* due to its clinical interest [12]. Fosfomycin is a broad-spectrum antibiotic effective against both Gram-positive and Gram-negative bacteria. It is primarily used to treat gastrointestinal and lower urinary tract infections but also in the treatment of various multidrug-resistant infections [13,14]. In *E. coli*, fosfomycin enters the cell through two transporters that import sugar phosphates in exchange for inorganic phosphate (Pi): the glycerol-3-phosphate transporter (GlpT) and the glucose-6-phosphate transporter (UhpT); their expression requires both the presence of their substrates and the starvation signaling molecule cAMP [15]. Once inside, fosfomycin acts as a phosphoenolpyruvate analog, irreversibly binding to enolpyruvoyl transferase MurA, an essential enzyme catalyzing the initial step of peptidoglycan biosynthesis, which ultimately results in cell lysis [16]. While spontaneous resistance can arise from mutations that modify or overexpress MurA [17,18], it is much more likely to occur through mutations that either inactivate GlpT or UhpT or suppress their expression by altering key genes involved in cAMP synthesis [15,19].

In contrast, *P. aeruginosa* lacks a functional *uhpT* homolog, leaving GlpT as the sole entry route for fosfomycin into the cells (protein identity, 83.3%) [20]. Moreover, cAMP signaling in this species is of less importance in regulating the uptake and metabolism of alternative carbon sources [21]. As a consequence, spontaneous fosfomycin resistance in *P. aeruginosa* is largely limited to arising from inactivating mutations in *glpT*, as has been demonstrated experimentally [22]. This specificity, combined with the loss-of-function nature of the mutations involved, makes fosfomycin resistance an ideal marker system for evaluating both spontaneous and induced mutation rates and spectra [23,24], adding to the relatively limited toolset available for investigating the molecular mechanisms underlying mutational processes in pseudomonads [25]. Additionally, this specificity can facilitate efficient screening schemes in genetic engineering. For instance, it can be used to either select desired mutants (e.g., by using the *glpT* locus as an integration site) or to counter-select undesired ones (e.g., in a *glpT* mutant, helping eliminate suicide vectors integrated into the genome) [26].

In principle, the above considerations should also apply to *P. entomophila*: no functional *uhpT* homolog is present in the genome, and there is no evidence that cAMP signaling differs qualitatively from what is known in *P. aeruginosa*. Motivated by these similarities, here we investigated the extent to which the specificity of spontaneous fosfomycin resistance observed in *P. aeruginosa* is conserved in *P. entomophila*. By characterizing independent fosfomycin-resistant mutants, we isolated resistant variants lacking alterations in *glpT* or any of its known regulatory elements. Experimental and computational analyses revealed a novel fosfomycin resistance mechanism in *P. entomophila*, arising from the placement of GlpT under the control of the Pho regulon. Finally, we used comparative genomics to show that this configuration is atypical across the diversity of pseudomonads, likely reflecting the specific demands and opportunities of *P. entomophila*’s habitat and lifestyle.

## 2. Results and Discussion

### 2.1. Fosfomycin Resistance in P. entomophila Involves Mechanisms Beyond glpT Inactivating Mutations

To elucidate whether the *glpT* gene is the primary target for fosfomycin resistance in *P. entomophila*, we began by conducting a standard Luria–Delbrück fluctuation assay to isolate 16 independent mutants resistant to 128 µg/mL of fosfomycin (Section 3). This concentration, a 32-fold increase over the minimal inhibitory concentration (MIC) for the wild type, was selected based on its use in previous *P. aeruginosa* studies [22,23]. From each isolated resistant mutant, we extracted chromosomal DNA, PCR-amplified their *glpT* sequences, and subjected them to Sanger sequencing (Section 3). Analysis of these sequences revealed that the majority (14/16, 87.5%) contained changes in the *glpT* gene, most likely representing loss-of-function mutations. This is best illustrated by the fact that out of these 14 mutations, 10 involved changes forcibly leading to substantial alterations in the resulting protein, such as insertions, deletions, or premature stop codons (Figure 1). In addition, 10/11 of the unique mutations (90.9%) mapped to transmembrane domains, representing a significant deviation from expectation since these regions account for only 29.7% of the protein sequence (binomial test, *p* = 1.3 × 10^−4^). This observation, along with the fact that these domains are known to be highly susceptible to destabilization by random mutations [27], further supports the idea that the adaptive value of these mutations lies in their likely loss-of-function nature. We also noted the presence of a prominent hotspot—absent in *P. aeruginosa*—that consisted of a four-base motif (GTGC) repeated three times consecutively, starting at base position 1268. Expansions and contractions of this motif caused a shift of the reading frame and were detected in five of the independently isolated mutants. Table 1 details all detected mutations in the *glpT* gene, including their position, molecular nature, and the corresponding amino acid and functional changes.

Next, we focused on the two mutants with no alterations along the *glpT* open reading frame, for which we followed a two-pronged approach (Section 3). On the one hand, we Sanger-sequenced the region approximately 100 bp upstream of *glpT* to search for mutations in the promoter region that might be altering gene expression. Additionally, we PCR-amplified and Sanger-sequenced the gene encoding GlpR, the local repressor of the *glp* regulon [28]. This regulon includes five operons involved in the uptake and metabolism of glycerol, including the *glpT*Q operon [29]. GlpR strongly inhibits these operons in the presence of glucose, and even in its absence, it continues to repress them unless sufficient levels of the inducers glycerol or glycerol-3-phosphate are present [30]. Our analyses revealed no alterations in any of these regions. Consequently, we subjected the two mutants to whole-genome sequencing and identified two unique mutations in *phoU*, a gene encoding a key negative regulator of the Pho regulon [31]. Both mutations likely result in loss of protein function: one involved a C → T substitution at position 406, leading to a premature stop codon (Q136*), and the other was a 519 bp deletion starting at position 302 (Figure 1). In *E. coli*, the Pho regulon includes at least eight operons that are co-regulated upon phosphate starvation, with their products involved in the uptake, transport, and metabolism of various phosphate sources [32]. Of note, *glpT* is not described as part of this regulon in either *E. coli* or *P. aeruginosa*.

### 2.2. Mutations in phoU Confer Fosfomycin Resistance in P. entomophila by Reducing GlpT Expression

Since the only two instances of fosfomycin resistance not involving changes in *glpT* pointed at the same regulatory gene, *phoU*, we hypothesized that either *glpT* or *glpR* were under the transcriptional control of the Pho regulon. The regulatory hub of the Pho regulon is the two-component signaling system PhoR/PhoB [33]. When phosphate is limited, the sensor histidine kinase PhoR acts as a kinase, transferring a phosphate group to the response regulator PhoB. Once phosphorylated, PhoB can bind to specific promoter regions, known as Pho boxes, activating the expression of the associated genes and operons [34]. However, when phosphate is abundant, PhoR primarily acts as a phosphatase, dephosphorylating PhoB and thereby preventing its role as a transcriptional activator [35]. PhoU is believed to assist PhoR in sensing the cell’s phosphate status and, depending on the availability of other micronutrients, to define PhoR’s role as a kinase or a phosphatase, and ultimately the expression of the Pho regulon genes [31,33].

To explore the possibility that the Pho regulon in *P. entomophila* may include either *glpT* or *glpR*, we first turned to the widely utilized promoter prediction software BPROM to search for Pho boxes in the promoter regions of these genes (Section 3). While no hits were found for *glpR*, we identified a potential PhoB binding site 76 base pairs upstream of the start of *glpT*’s open reading frame: an ATAAAA sequence that is conserved and experimentally validated in the Pho regulon members *pstS* and *phoE* of *E. coli* [36]. To establish the functional significance of this finding, we next performed real-time PCR to quantify *glpT* expression levels in both *phoU* mutant backgrounds (Section 3). Figure 2A confirms that, indeed, *glpT* transcript levels were strongly reduced in these *phoU* mutants, exhibiting a nearly 20-fold decrease in expression compared to the wild-type.

We finally sought to quantify the extent to which the drastic reductions in *glpT* expression translate into increases in fosfomycin resistance. To this end, we used the standard broth micro-dilution method to estimate the MIC for both *phoU* mutants (Section 3). Besides the wild-type, we also included as a reference a *glpT* mutant with a nonsense mutation (Q372*). This mutant should represent the maximum resistance level achievable when no fosfomycin is transported via GlpT, a situation where all entry is expected to be limited to inefficient passive diffusion through the outer membrane porins [37]. Previous studies on fosfomycin-resistant *P. aeruginosa* mutants with various *glpT* mutations showed that nonsense, missense, and indel mutations can all lead to large elevations of the MIC, with just minor differences among these classes [22]. Figure 2B demonstrates that the *phoU* mutants indeed exhibited a substantial increase in resistance levels (MIC: 256 µg/mL, a 64-fold elevation over the wild type). However, this increase was of moderate magnitude compared to the presumed upper bound observed in the *glpT* loss-of-function mutant (MIC: 4096 µg/mL, a 1024-fold increase over the wild type), suggesting that some fosfomycin still enters the cell via GlpT, despite the reduced expression measured in the *phoU* mutants.

### 2.3. Control of GlpT by the Pho Regulon Seems Peculiar to P. entomophila

Our results indicate that, unlike in *E. coli* and *P. aeruginosa*, glpT is regulated by the Pho regulon in *P. entomophila*. At first glance, integrating a sugar-phosphate transporter into a phosphate starvation regulatory network may seem appropriate since the global logic of the Pho regulon is to enable the hierarchical and orderly activation of measures to acquire phosphorus from increasingly desperate sources [38]. This reasoning raises the question of why this regulatory arrangement is not observed in *E. coli* and, most intriguingly, in the closely related *P. aeruginosa*. In other words, if we look across the diversity of *Pseudomonas* species, is *P. entomophila* an outlier regarding *glpT* regulation, or is *P. aeruginosa* the exception?

To address these questions, we compared the *glpT* promoter sequences across 18 representative Pseudomonas species, as well as E. coli—a standard outgroup—and *Azotobacter vinelandii*, a prominent member of the *Pseudomonadaceae* family so closely related to *Pseudomonas* that it is sometimes classified within this genus [39]. To this end, we began by examining the evolutionary relationships among the 20 selected species, as illustrated in Figure 3, which presents a phylogenetic tree based on pairwise genome distances (Section 3). Next, we turned to the BPROM promoter prediction software to identify transcription factor binding sites within the promoter regions of the collection of *glpT* homologs. Figure 3 also displays the distribution of the detected sites among species, which, for reference, are organized according to the clustering suggested by the phylogenetic tree.

Three key observations emerge from this analysis. First, 8 out of the 20 species lacked a clear *glpT* homolog despite the use of complete and well-annotated genomes from the MicroScope database [40]. Second, there was substantial variation in the transcription factors predicted to regulate *glpT* homologs across the species. Among the 16 transcription factors identified, the majority (62.5%, 10/16) were unique to a single species, and the most common transcription factor, SoxS (a transcriptional activator of the superoxide response regulon [41]), was present in only six species. We note that these results likely underestimate the true variability since BPROM has a recognition sensitivity of 83%, which is probably a compromise to minimize false positives [42]. Finally, a PhoB binding site was detected exclusively in *P. entomophila*, indicating that the control of GlpT by the Pho regulon is, if not unique, at least highly specific to this bacterium. Taken together, these observations suggest that the availability of external glycerol-3-phosphate varies widely across environments, a heterogeneity manifesting in at least two dimensions: first, in terms of absolute abundance, which sometimes can be virtually zero, as suggested by species lacking the transporter altogether; and second, in terms of the timing and environmental conditions associated with its presence, as suggested by the highly diverse and poorly conserved regulatory mechanisms across species.

In the specific case of *P. entomophila*, the suggestion would be that its ecology often features conditions where glycerol-3-phosphate is not only abundant but also frequently coincides with phosphate starvation; so it has become evolutionary worth it to integrate GlpT under the Pho regulon to utilize glycerol-3-phosphate as both a carbon and phosphate source. Of note, glycerol-3-phosphate is actively generated in insect flight muscles to facilitate NADH transport into mitochondria for oxidative phosphorylation [43], and its accumulation has been observed in butterfly larvae, where it likely serves as an energy source during this specific developmental stage [44]. Although *P. entomophila* is considered an opportunistic entomopathogen, and the extent of time it spends within hosts versus a free-living bacterium is uncertain [3], it is intriguing to speculate whether the availability of glycerol-3-phosphate in insects might have driven the integration of GlpT under the Pho regulon.

At a broader scale, these results highlight how readily regulatory evolution can lead to the rapid divergence of resistance mechanisms, even among closely related species. Previous studies have documented that species within the same order, such as *E. coli*, *Salmonella enterica*, and *Yersinia pestis*, can drastically differ in their responsiveness to environmental cues that activate resistance to the antimicrobial agent polymyxin B [45,46,47]. In contrast to our study, in these cases, the differences emerged by either the presence or absence of a key regulatory protein or by substantial molecular divergence in this protein rather than changes in promoter regions. Yet, the evolution of promoter regions, including the gain and loss of binding sites, is well-documented in both eukaryotes and prokaryotes and is believed to be a mechanism favored over changes in regulatory proteins due to its higher specificity [48,49]. Moreover, in bacteria, promoter evolution such as the one described here has been shown to occur readily through horizontal gene transfer [50,51]. In summary, our results contribute to the literature showing that regulatory circuits are in a constant state of evolution, underscoring the remarkable ability of organisms to adapt to new niches with different demands and varying stimuli patterns.

## 3. Materials and Methods

### 3.1. Isolation of Spontaneous Fosfomycin Resistant Mutants

Fifteen parallel cultures of *P. entomophila* L48 (obtained from Jens Rolff’s laboratory [52]) were inoculated in 3 mL of Luria broth (LB; Miller’s modification: 10 g/L of sodium chloride, 10 g/L of tryptone, 5 g/L of yeast extract; purchased in powder formulation from Condalab, Madrid, Spain) in 15 mL Falcon tubes and incubated overnight at 28 °C with shaking at 180 rpm for 48 h. The next day, 100 µL of each culture were plated onto LB-agar Petri dishes containing 128 µg/mL of fosfomycin (purchased as a disodium salt, IM, from Laboratorios Ern, Barcelona, Spain). After overnight incubation at 28 °C without shaking, a single colony from each fosfomycin plate was selected at random and re-streaked for purification under the same conditions. After further incubation, a single colony from each of these re-streaked plates was collected and stored at −80 °C for further analysis.

### 3.2. Antimicrobial Susceptibility Testing

The standard broth microdilution method was used to estimate the minimum inhibitory concentration (MIC) for both the wild-type and mutant strains. We followed the Clinical and Laboratory Standards Institute [53] guidelines, with the exception of using Luria broth (LB) instead of Mueller–Hinton. Briefly, strains were retrieved from −80 °C storage and inoculated into 3 mL of LB in 15 mL Falcon tubes. These cultures were incubated overnight at 28 °C with shaking at 180 rpm. After two overnight incubations, cultures were centrifuged and adjusted in saline solution to a 0.5 McFarland standard, corresponding to 1.14 ± 0.4 × 10^7^ CFU/mL (mean ± standard deviation, *n* = 3). Note that this value is lower than what is usually assumed for *E. coli* [54]. The cultures were then diluted at 1:150, and 100 µL aliquots were inoculated into microtiter plates containing a two-fold concentration gradient of fosfomycin in a final volume of 200 µL. The MIC was defined as the lowest fosfomycin concentration that completely prevented turbidity from forming after 48 h of static incubation at 28 °C.

### 3.3. Sanger and Whole-Genome Sequencing

Strains were retrieved from −80 °C storage and streaked on LB agar containing 128 µg/mL of fosfomycin. Genomic DNA was extracted using the DNeasy UltraClean Microbial Kit (Qiagen, Hilden, Germany) following manufacturer instructions. For Sanger sequencing, the *glpT* open reading frame was PCR amplified and sequenced using the forward primer 5′-CAA CAG GCC TGG GCA ATA-3′ and reverse primer 5′-CTG AAA ACA CAA AGC CCG G-3′. The *glpT* promoter region was amplified and sequenced using an alternative forward primer, 5′-GCT GAC GAT CCA GGT TGA CT-3′, which binds 142–123 bp upstream of *glpT*. The *glpR* open reading frame was amplified using forward primer 5′-AGC GCA ACA GAA AGC CTG TC-3′ and reverse primer 5′-GGG AAG ATC ACG ACG CCT GAT A-3′. PCR amplicons were purified using the DNeasy UltraClean Microbial Kit (Qiagen), and Sanger sequencing was conducted by Macrogen Inc. (Seoul, Republic of Korea). Mutations were identified by aligning the resultant sequences against the corresponding sequences extracted from the reference genome (downloaded from the MaGe Platform (https://mage.genoscope.cns.fr/ (accessed on 15 August 2024) [40]). These alignments were conducted using Vector NTI Advance™ 11.0 (Invitrogen, Waltham, MA, USA). For whole-genome sequencing, extracted genomic DNA was quantified using the Qubit 4 Fluorometer (Invitrogen, USA). Illumina paired-end sequencing was performed by Novogene (Cambridge, UK) at >100× coverage on the two isolates with no detected mutations in *glpT*, along with the isolate containing the Y407* mutation in *glpT*, as a control. Alignment and analysis were conducted using the computational pipeline breseq [55], using the reference genome retrieved from the MicroScope database. The raw reads were deposited in the NCBI Sequence Read Archive (PRJNA1174203).

### 3.4. Quantitative Real-Time PCR

RT-qPCR was performed to measure *glpT* gene expression using the comparative ΔΔCt method, with the ribosomal gene *rpsL* as the endogenous control [56,57]. Strains were retrieved from −80 °C storage and *grown in 1.5 mL of Luria broth* at 28 °C until the mid-log phase (OD600~0.5). After centrifugation, cells were resuspended in 100 µL of freshly prepared TE buffer (10 mM Tris-HCl, pH 8.0, 1 mM EDTA) containing 0.4 mg/mL of lysozyme (Thermo Fisher Scientific, Alcobendas, Spain) to facilitate cell lysis. RNA was extracted using the GeneJET RNA Purification Kit (Thermo Fisher Scientific, Spain) according to the manufacturer’s protocol. Extractions were performed in biological and technical triplicates. RNA concentration and purity (A260/280) were assessed using a NanoDrop^®^ ND-1000 Spectrophotometer (NanoDrop Technologies Inc., Montchanin, DE, USA). To eliminate residual DNA, RNA samples were treated with DNase I (Thermo Fisher Scientific, Spain) before reverse transcription. cDNA was synthesized using the High-Capacity cDNA Reverse Transcription Kit (Thermo Fisher Scientific, Spain) with random hexamers, including RNase Inhibitor to prevent degradation. The qPCR reactions were conducted in triplicate using SYBR Green PCR Master Mix (Thermo Fisher Scientific, Spain) on a LightCycler^®^ 480 system in a final volume of 20 µL per well, using alphanumerically coded white 96-well plates (Roche, Basel, Switzerland). For *glpT*, the following primers were used: forward, 5′-CGC CGG TCG AGC AAT ACA A-3′; reverse: 5′-CTC GTA GAA GAA GTA CGC CCA C-3′. For *rpsL:* forward, 5′-CCT AAC TCG GCA CTG CGT AA-3′; reverse, 5′-TGG TAA CGA ACA CCT GGC AA-3′.

### 3.5. Computational Methods

The genome sequences of *P. entomophila L48* and all the other 19 species were retrieved from the MaGe Platform (https://mage.genoscope.cns.fr/, accessed on 15 August 2024). A list with the details of these genomes is presented in Appendix A. To build the phylogenetic tree, we resorted to MaGe’s online genome clustering tool, which computes pairwise genome distances using the software Mash [58]. These distances (D) strongly correlate with the Average Nucleotide Identity (ANI), such as D ≈ 1 − ANI. Using all the pairwise distances, a tree was built dynamically using the neighbor-joining Javascript package [59]. The tree was saved in the Newick format, and visualization was conducted using the R phylogenetics package “ape” [60]. To identify putative transcription factor binding sites, the 400 bp upstream of the start site of the gene of interest (*glpT* or *glpR*) were submitted in fasta format to the promoter prediction tool BPROM (Softberry Inc., Mt Kisco, NY, USA) [42], freely available online from www.softberry.com/berry.phtml?topic=bprom&group=programs&subgroup=gfindb (accessed on 15 August 2024).

## 4. Conclusions

Here, we report that the single-locus specificity that spontaneous fosfomycin resistance shows in *P. aeruginosa* is not conserved in *P. entomophila*. While inactivating mutations in glpT remain the most common molecular mechanism underpinning resistance, we observed a sizable fraction of spontaneous mutants with alterations in phoU, a key regulatory element of Pho regulon. Computational evidence indicates that the observed link between phoU and glpT expression levels may be attributed to a predicted binding site for the transcription factor PhoB in the promoter region of glpT. Further research is required to elucidate the precise molecular details of this predicted interaction. Anyway, given the substantial increases in resistance conferred by any of these mutations, spontaneous fosfomycin resistance can be a valuable marker for studying mutation rates in *P. entomophila*. Likewise, these substantial increases in resistance can make it useful for selection and counterselection in genetic engineering. However, its effectiveness as a marker for studying mutation spectra in this species is more limited than in *P. aeruginosa*, as the involvement of multiple genes imposes a larger workload. Finally, our analyses indicate that glpT regulation by the Pho regulon is likely specific to the habitat and lifestyle of *P. entomophila*, being uncommon among other pseudomonads. This finding highlights how ecological differences can drive the rapid divergence of resistance mechanisms and suggests that future research may uncover diverse and disparate genetic elements underlying spontaneous antibiotic resistance, even among related species.

## Figures and Tables

**Figure 1 antibiotics-13-01008-f001:**
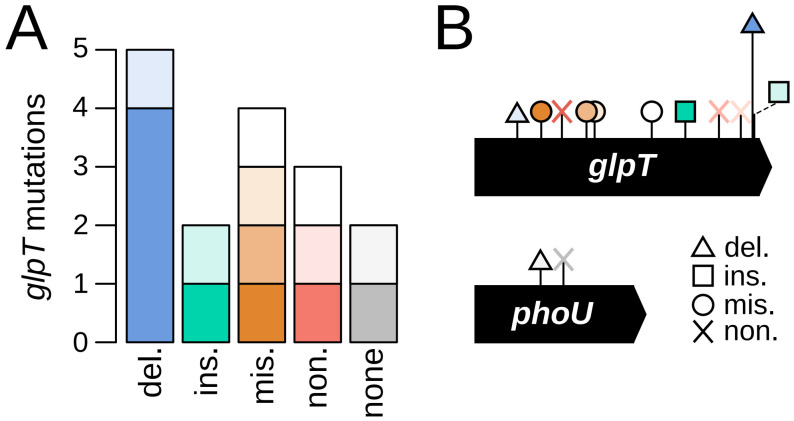
Genetic features of spontaneous fosfomycin-resistant *P. entomophila* isolates: (**A**) Distribution of mutation types as detailed in Table 1, categorized by deletions, insertions, missense, nonsense, and absence of mutations. Different shades of the same color represent distinct mutations within the same category; (**B**) Mutation mapping along the open reading frames of the *glpT* and *phoU* genes. Bar height indicates the abundance of mutations at each position. As shown in Table 1, all mutations occur only once, except the deletion at codon 423 (which appears five times. Note that mutations are scattered along the *glpT* sequence, consistent with their presumed loss-of-function nature. The colors correspond to those in panel (**A**).

**Figure 2 antibiotics-13-01008-f002:**
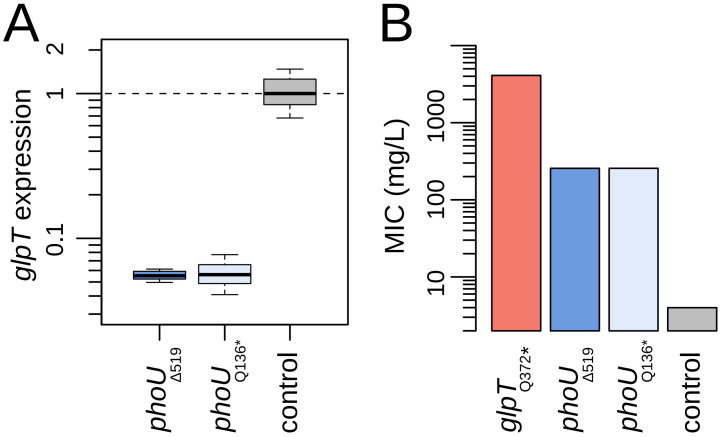
Phenotypic features of spontaneous fosfomycin-resistant *P. entomophila* isolates: (**A**) Relative expression levels of the *glpT* gene in the two mutants with unique loss-of-function mutations in phoU and the wild-type strain, as measured by qRT-PCR. Colors correspond to those in Figure 1A; (**B**) Minimum inhibitory concentration (MIC) values in the same two *phoU* mutants and the wild-type strain (average of three replicates, with no observed deviation among them). For reference, a *glpT* mutant with a nonsense mutation (Q372*) is included, representing the upper bound for resistance when GlpT-mediated fosfomycin transport is absent. Note the y-axis logarithmic scale. The colors correspond to those in Figure 1A. Symbols follow standard convention: deletion (Δ), stop codon (*).

**Figure 3 antibiotics-13-01008-f003:**
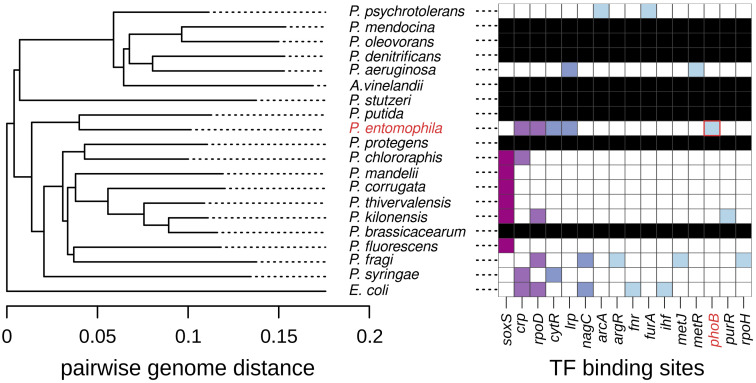
Computational analysis of *glpT* regulation across pseudomonads and related species: the neighbor-joining phylogenetic tree on the (**left**) is based on pairwise distances among the genomes analyzed in this study (Appendix A). The heatmap on the (**right**) illustrates the abundance and distribution of transcription factor binding sites predicted upstream of the *glpT* gene across the analyzed species. Black rows indicate species lacking a clear *glpT* homolog. The colors indicate the frequency of each binding site across species, with dark purple representing the highest occurrence, light blue representing a single occurrence, and white representing no hits. The labels of *P. entomophila* and *phoB* are highlighted in red for reference.

**Table 1 antibiotics-13-01008-t001:** Sanger sequencing results of the *glpT* gene in spontaneous fosfomycin-resistant isolates of *P. entomophila*. Symbols follow standard convention: deletion (Δ), insertion (+), stop codon (*).

Position	Base Change	Codon	*glpT* Mutations	Effect	Domain
196	Δ14 bp	65	Q65 Δ14 bp	frameshift	transmembrane
305	t → g	103	L102R	missense	transmembrane
399	g → a	133	W133*	nonsense	transmembrane
511	g → a	171	G171D	missense	transmembrane
549	g → t	183	W183C	missense	transmembrane
808	c → a	270	R270S	missense	transmembrane
961	+4 bp	321	N321 + 4 bp	frameshift	Intracellular loop
1114	c → t	372	Q372*	nonsense	transmembrane
1221	c → a	407	Y407*	nonsense	transmembrane
1268	+4 bp	423	S423 + 4 bp	frameshift	transmembrane
1268	Δ4 bp	423	S423 Δ4 bp	frameshift	transmembrane
1268	Δ4 bp	423	S423 Δ4 bp	frameshift	transmembrane
1268	Δ4 bp	423	S423 Δ4 bp	frameshift	transmembrane
1268	Δ4 bp	423	S423 Δ4 bp	frameshift	transmembrane
-	-	-	none	-	-
-	-	-	none	-	-

## Data Availability

All data presented in this study are contained within the article or Appendix A. Raw whole-genome sequences are deposited in the NCBI Sequence Read Archive (PRJNA1174203).

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
