# Peer review of "Novel Fosfomycin Resistance Mechanism in *Pseudomonas entomophila* Due to Atypical Pho Regulon Control of GlpT"

_antibiotics, 2024, doi:10.3390/antibiotics13111008_

Round 1

Reviewer 1 Report

Comments and Suggestions for Authors

1. The work is very interesting to the relevant sciences.

However, massive improvement must be done in the text. In particular, i have found a lot of sentences started with We.

lines 95-131, etc. therefore, the whole text should be considered or revised.

2. Authors should make a Table and explain the fifteen mutants position (DNA sequence) of each mutants  P. aeruginosa. explain how these mutations affects the strains roles in fosfomycin resistance mechanism in Pseudomonas entomophila.

3. Massive language edition is required is the whole text, it seems the texts are made using AI. please rewrite to make a clear flow of the study. 

Comments on the Quality of English Language

substantial level editing is required in the whole text. should be edited by a certified professional editor to make it sound

Author Response

Please see the attachment (pages 2-3)

Reviewer 2 Report

Comments and Suggestions for Authors

In this study, Sánchez-Maroto et al. worked to test a GlpT-based resistance mechanism in Pseudomonas entomophila. Although the study is well formulated, the following points need to be addressed.

Points of major concerns:

1.    The authors proposed that the 9 mutations were associated with alterations in the corresponding protein. How did they build this assumption? There are no experiments of the resultant proteins.

2.    Figure 1 makes me confused. The authors reported that no glpT mutations were detected in the WT strain, however in the graph, there is still mutations within the WT bar!

3.    I am wondering, why did the authors looked only for a region of 100bp upstream of glpT during investigating the promoter, which could normally include 100-1000 bp?!

4.    P. entomophila is known to express fosA which is regarded as fosfomycin resistant gene. It is surprising for me that authors did not investigate whether there is a connection between pho and/or glp and this gene.

Points of minor concerns:

1.    I find it somehow strange, to use the 125 ug/ml of fosfomycin and justifying this by founding it in a study performed in P. aeuroginosa. The authors need to do Luria-Delbrück fluctuation assays using data based on studies performed in P. entomophila.

2.    Please indicate in table 1 that the stars refer to the two strains with nonsense mutations which used later.

3.    The authors need to include a diagram indicating the component of glpT and phoU loci.

4.    In figure two, how many replicates were used to measure the MIC in graph 2, B?

5.    In line 294, the authored mentioned that the 0,5 McFarland is equal to more or less 1 x 108 CFU/ml in E. coli. They can not use this value for the P. entomophila. This will raise the question again what was the CFU values used in these experiments?

Comments on the Quality of English Language

The quality of writing is average and could be followed.

Author Response

Please see the attachment (pages 4-7)

Reviewer 3 Report

Comments and Suggestions for Authors

Reviewer’s comments

The study investigates fosfomycin resistance in Pseudomonas entomophila, comparing it to Pseudomonas aeruginosa. The authors show resistance in P. aeruginosa is linked to mutations in the glycerol-3-phosphate transporter (GlpT), some P. entomophila mutants had mutations in the phoU gene, reducing glpT expression. The study is not very well planned, the experiments performed are not enough to conclude the said findings. Here are my comments:

1.     2.1. Fosfomycin resistance in P. entomophila involves mechanisms beyond glpT inactivating mutations: The authors studied these mechanisms on the spontaneous resistant mutants of Fosfomycin. This study need to be performed on the resistant mutants passaged multiple times and then passaged at least for 7 days in the drug free media to know if these are stable mutants. Then these types of conclusions can be derived.

2.     The heading 2.1 and 2.2 are same.

3.     Figure 1B does not explains properly. The authors could use a different graph such as bar graph with the axis plotting the frequency of mutations.

4.     Figure 2B, the Y-axis labels are missing, therefore it is very difficult to see the MIC values. Please label the axis properly.

Comments on the Quality of English Language

The quality of english is moderate and can be improved.

Author Response

Please see the attachment (pages 8-9)

Reviewer 4 Report

Comments and Suggestions for Authors

Pseudomonas entomophila is a widespread bacterium capable of killing insects across 8 orders, making it a model for host-pathogen studies and a promising tool for biological pest control. Laura et al. identified 15 fosfomycin-resistant mutants, two of which had mutation in phoU, a regulator of the phosphate starvation (Pho) regulon. The data revealed an atypical Pho regulon control of glpT, suggesting that this regulation is likely specific to the habitat and lifestyle of P. entomophila, a configuration uncommon among other pseudomonads. This highlights how ecological differences can drive rapid divergence in resistance mechanisms, indicating that spontaneous antibiotic resistance may cause by diverse genetic elements.

I have several comments according to this manuscript:

Laura et al. gave a novel idea that the same spontaneous antibiotic resistance can lead by different genetic changes. This is also excellent combination of experimental data and computational data. However, more solid experimental evidence should be provided to support this novel atypical regulation pathway, especially the computational analysis indicates that phoB can bind to glpT promoter.

Majors:

1.     In figure 1 panel B, author should map those mutations onto the GlpT structural model, this may make the figure more informative and easier for reader to understand where those mutations are.

2.     In figure 2 panel A, author presented the glpT expression level among phoU mutant and wildtype. However, phoB is the transcription factor which directly interact with glpT promoter. Author should include phoB mutant or knockout samples into the glpT expression level experiments, especially author also found out that there is a phoB binding site within the glpT promoter sequence. This might help to confirm that GlpT is control by Pho regulon in P. entomophila.

3.     Figure 2 panel B should include the error bar and indicate the repeat times of the MIC experiment.

Minors:

1.     The subtitle 2.1 and 2.2 are identical, please combine these two sections or give a new subtitle.

2.     Figure 3 should indicate panel a and b corresponding to the figure legend.

3.     In line 78, author mentioned that there is no functional uhpT homologue in P. entomophila, are there any glpT homologue exist in P. entomophila? If so, what is the protein sequence identity?

4.     In the discussion, author should discuss the relationship between the atypical GlpT regulation and the living environment and lifestyle of P. entomophila. What makes P. entomophila adopts this special GlpT regulation pathway.

Author Response

Please see the attachment (pages 10-14)

Round 2

Reviewer 1 Report

Comments and Suggestions for Authors

authors improved the manuscript as per comments. it can be accepted

Reviewer 2 Report

Comments and Suggestions for Authors

The authors investigated and answered all relevant questions.

Comments on the Quality of English Language

The english quality is now better and could be followed without great issues.

Reviewer 4 Report

Comments and Suggestions for Authors

The authors have addressed all the comments clearly. Well done!